# Minimal Contact Robotic Stroke Rehabilitation on Risk of COVID-19, Work Efficiency and Sensorimotor Function

**DOI:** 10.3390/healthcare10040691

**Published:** 2022-04-06

**Authors:** Bu Hyun Yoon, Chanhee Park, Joshua (Sung) Hyun You

**Affiliations:** 1Sports Movement Artificial-Intelligence Robotics Technology (SMART) Institute, Department of Physical Therapy, Yonsei University, Wonju 26493, Korea; ybh54888@naver.com (B.H.Y.); chaneesm@gmail.com (C.P.); 2Department of Physical Therapy, Yonsei University, Wonju 26493, Korea

**Keywords:** walkbot, robotic-assisted gait training, work efficiency, COVID-19

## Abstract

Patients with hemiparetic stroke undergo direct, labor-intensive hands-on conventional physical therapy to improve sensorimotor function, spasticity, balance, trunk stability, and activities of daily living (ADLs). Currently, direct, intensive hands-on therapeutic modalities have increased concerns during the coronavirus (COVID-19) global pandemic. We developed an innovative Walkbot to mitigate the issues surrounding conventional hands-on physical therapy. We aimed to compare the effects of minimal-contact robotic rehabilitation (MRR) and full-contact conventional rehabilitation (FCR) on static and dynamic balance, trunk stability, ADLs, spasticity, and cognition changes in patients with hemiparetic stroke. A total of 64 patients with hemiparetic stroke (mean age = 66.38 ± 13.17; 27 women) underwent either MRR or FCR three times/week for 6 weeks. Clinical outcome measurements included the Trunk Impairment Scale (TIS), the Berg Balance Scale (BBS), the modified Ashworth Scale (MAS), the Fugl—Meyer Assessment (FMA), and the modified Barthel Index (MBI) scores. A 2 × 2 repeated analysis of variance (ANOVA) was performed, and an independent *t*-test was used to determine statistical differences in the physiotherapists’ work efficiency and COVID-19 transmission risk. The ANOVA showed that MRR had effects superior to those of FCR on the TIS, the BBS, the FMA, and the MBI (*p* < 0.05), but not on the MAS (*p* = 0.230). MRR showed a greater decrease on the physiotherapist’s work efficiency and COVID-19 transmission risk (*p* < 0.05). Our results provide clinical evidence that robot-assisted locomotor training helps maximize the recovery of sensorimotor function, abnormal synergy, balance, ADLs, and trunk stability, and facilitates a safer environment and less labor demand than conventional stroke rehabilitation.

## 1. Introduction

Robotic-assisted gait training (RAGT) is a promising rehabilitation technology for locomotor training in patients with stroke. This modality is important as it encourages hands-on contact and mitigates transmission concerns during the global COVID-19 pandemic [1,2]. Patients with stroke exhibit sensorimotor dysfunction and spasticity, which often affect trunk stability, balance, and activities of daily living (ADLs) [3,4,5,6,7,8]. Currently, neurorehabilitation approaches involve direct, labor-intensive hands-on therapy, while outcome measure studies show variable results [9]. Accumulative evidence on RAGT suggests its promising results for trunk stability, balance, spasticity, abnormal synergistic patterns, and ADLs when combined with conventional physical therapy [10,11,12,13,14]. Additionally, neurodevelopmental treatment (NDT)-based conventional physical therapy presents practical issues associated with its labor intensiveness, increased physical stress on therapists, risk of falls, and lack of repetition to generate sufficient neuroplasticity [15]. However, the main issues with direct, labor-intensive hands-on therapy have become more critical during the COVID-19 pandemic [16,17]. There is a clear need to increase the effectiveness and sustainability of robotic intervention strategies while minimizing or eliminating physical contact to prevent airborne and other modes of COVID-19 transmission during stroke rehabilitation.

To mitigate the inherent problems of conventional physical therapy during the COVID-19 pandemic, we developed an innovative Walkbot RAGT with minimal or no physical contact to improve balance, trunk stability, and ADLs [12,18]. RAGT comprises intensive, repetitive gait cycles using body-weight support to strengthen weak lower limbs while meeting the musculoskeletal requirements for normal gait. Moreover, RAGT exerts less cardiorespiratory stress than overground gait training or gait training without robotic assistance [19]. The Walkbot system is designed to provide intensive training with minimal or no physical contact or stress to the therapist, decrease the risk of falling, and provide an ample amount of repetition [12].

Despite such potential clinical advantages, clinical outcomes, the physiotherapists’ work efficiency, and lower transmission risk of COVID-19, minimal-contact robotic rehabilitation (MRR) has not been investigated in hemiparetic stroke rehabilitation. Therefore, the primary objective of the present study was to compare the risk of COVID-19 vulnerability, including the transmission risk of COVID-19 (social distancing, duration of contact time, and evidence of COVID-19 transmission) and physiotherapists’ work efficiency (labor intensiveness, physical stress, or level of effort) in MRR and full-contact conventional rehabilitation (FCR). The secondary objective was to ascertain the comparative effects of MRR and FCR on static and dynamic balance, trunk stability, ADLs, and spasticity in a large sample of patients with hemiparetic stroke. We hypothesized that MRR would improve clinical indices of the risk of COVID-19 vulnerability, the Trunk Impairment Scale (TIS), the Berg Balance Scale (BBS), the modified Ashworth Scale (MAS), the Fugl–Meyer assessment (FMA), and the modified Barthel Index (MBI), when compared to those of FCR conventional physiotherapy.

## 2. Materials and Methods

### 2.1. Participants

A convenience sample of 64 patients with stroke (mean age = 66.38 ± 13.17; 27 women) admitted from July 2019 to December 2021 were prospectively evaluated and the data stored in the Clinical Data Warehouse (CDW) in the Chungdam hospital (Seoul, Republic of Korea), including a database of electronic medical records obtained from inpatients for further analysis. The CDW contains nearly all patient medical records, including every field note written by the medical staff (admission and discharge notes, progress reports, and nursing data), patient information data and records obtained (insurance, diagnostic codes, age, sex, and vital signs), test results (laboratory tests, functional assessments, and imaging studies), and treatment modalities used (medications, therapies, and medical procedures). Data were de-identified and provided to the research team. Consequently, patient consent was not required. This study was approved by the hospital’s internal review board (IRB No. CDIRB-2021-01). The inclusion criteria were as follows: (1) history of cortical/subcortical ischemic stroke, (2) age between 18 and 99 years, (3) clinical presentation of first or prior stroke with no residual deficits affecting ambulation, (4) ability to follow a two-step command, (5) suitability for gait training as assessed clinically by the ability to ambulate at least one step with a device and/or assistance, (6) height: 132–200 cm, (7) hip-knee joint length: 33–48 cm, and (8) knee joint–foot length: 33–48 cm. The exclusion criteria were as follows: (1) history of cerebellar/brainstem stroke; (2) body weight >135 kg; (3) uncontrolled hypertension (stage 2) with blood pressure >160/100 mmHg; (4) cardiopulmonary impairments affecting the ability to perform the ambulation test; (5) integumentary impairment such as skin breakdown or bedsores around the area where the suspension belt is placed; (6) significant and persistent mental illness; (7) a lower-extremity fixed contracture or deformity; (8) bone instability (non-consolidated fractures, unstable spinal column, or severe osteoporosis necessitating treatment with bisphosphonates); (9) other neurodegenerative disorders (amyotrophic lateral sclerosis and Parkinson’s disease); (10) a modified Ashworth scale score >3 in the affected leg; (11) significant back or leg pain resulting in an inability to tolerate movement; (12) decreased sensation impairing the ability to perceive whether the device is properly fitted; and (13) aphasia preventing the ability to communicate discomfort.

### 2.2. Clinical Testing Procedure

The present study used a two-group pretest and posttest design in which all patients completed the pretest, the intervention, and the posttest. Clinical outcome tests included standardized TIS, BBS, MBI, MAS, FMA, and a postquestionnaire to assess the physical therapists’ work efficiency (labor intensiveness, physical stress, and level of effort) and transmission risk of COVID-19 (social distancing, duration of contact time, and evidence of COVID-19 transmission).

#### 2.2.1. Trunk Impairment Scale

The TIS was used to determine intervention-related changes in static and dynamic sitting balance as well as selective movements of the trunk in a sitting position. The scoring system, which is composed of 3 subscales, has a total of 17 items with an achievable score from 0–23 [20]. Reliability was reported to be intraclass correlation coefficient (ICC)_3,k_ = 0.99 and *r* = 0.86, respectively [21].

#### 2.2.2. Berg Balance Scale

The BBS was used to measure performance-oriented balance in patients with stroke and balance impairment. The test comprises 14 balance-related tasks, ranging from standing up from a sitting position to standing on 1 foot. The degree of success in achieving each task was given a score of 0 (unable to perform) to 4 (able to perform independently); scores were summed to obtain the final score [22]. Reliability and validity were reported to be ICC_3,k_ = 0.95 and *r* = 0.93, respectively [23].

#### 2.2.3. Modified Barthel Index

The MBI is a widely used assessment tool that measures dependence levels in performing functional ADLs. It has 3 different categories on a 5-point scale as follows: a score of 0–5 for bathing, personal hygiene, and wheelchair management; 0–10 for feeding, dressing, toilet transferring, stair climbing, bladder control, and bowel control; and 0–15 for chair/bed transfers and ambulation. A score of 0–24 indicates total dependence; 25–49 indicates severe dependence; 50–74 indicates moderate dependence; 75–90 indicates mild dependence; 91–99 indicates minimal dependence; and 100 indicates independence [24]. Reliability and validity were reported to be ICC_3,k_ = 0.99 and *r* = 0.95, respectively [25].

#### 2.2.4. Modified Ashworth Scale

The MAS was used to determine the degree of spasticity during passive ankle dorsiflexion. The patient was comfortably seated in a chair, and the physiotherapist applied rapid, passive ankle dorsiflexion. The degree of the participant’s muscle response to the passive stretching was visually inspected. The ordinal grading scale ranged from 0 (“no increase in muscle tone”), 1 (“slight increase in tone followed by minimal resistance”), 2 (“more marked increase in tone but affected part easily moved”), 3 (“considerable increase in tone and passive movement difficult”), and 4 (“the affected part is rigid in flexion or extension”) [26]. Reliability was reported to be ICC_3,k_ = 0.92 [27].

#### 2.2.5. Fugl–Meyer Assessment

The FMA was used to examine lower extremity sensorimotor function and hip–knee–ankle joint function. The test consists of nine questions on reflex activity, supine position, volitional movement with synergy, volitional movement with mixing synergies, volitional movement with little or no synergy, and normal reflex activity. The ordinal grading scale scores obtained were 0 (“cannot perform”), 1 (“partially perform”), and 2 (“completely perform”). The scores ranged from 0–28 points [28]. Reliability and validity were reported to be *r* = 0.96 and ICC_3,k_ = 0.97, respectively [29].

#### 2.2.6. Postquestionnaire

The physiotherapist’s work efficiency questionnaire included questions on labor intensiveness, physical stress, and level of effort; the perceived transmission risk of COVID-19 section had questions on social distancing, duration of contact time, and evidence of COVID-19 transmission. The scale ranged from 0 (“labor free”) to 10 (“maximal labor required”). A total of 10 physiotherapists with 2 years of experience in both conventional physical therapy and robotic therapy participated in this survey.

### 2.3. Intervention

MRR was performed three times/week for 6 weeks (18 sessions total) with a duration of 30 min each on the Walkbot-G system (P&S Mechanics, Seoul, Korea), excluding the set-up time. Break time was provided when requested by the participant; however, intervention time was maintained for at least 30 min. The Walkbot-G system is an interactive robot-assisted locomotor training device with a built-in ankle actuator that provides an optimal ankle motion trajectory during ambulation. An adjustable leg length and control of the ankle joint range of motion enabled the Walkbot-G system to accurately approximate human kinematics and kinetics [30,31]. These data were then used to automatically adjust the length of the exoskeleton legs and optimal gait cycle according to each participant’s condition. Each participant then used a suspension vest secured with elastic straps that were connected to the harness mounted on the counterweight system. Depending on the participants’ initial clinical conditions (e.g., pain, muscle weakness, spasticity, tolerance, fatigue, or endurance), approximately 40–60% (adjustable range, 0–100%) of the total body weight was sustained in the first session and then gradually decreased in 5–10% increments per session. Furthermore, real-time audiovisual biofeedback concerning gait kinematics (joint angles) and kinetic forces (active, resistive torque, and stiffness) on the ankle, knee, and hip interlimb joint movement, as well as on the center of pressure from the force plate mounted on the treadmill, was obtained. During and after each session, the participants were provided with constant verbal encouragement using knowledge derived from real-time kinematic and kinetic data [11]. No safety issues were reported, and none of the participants experienced side effects during the MRR (Figure 1).

The FCR includes a manual therapeutic approach, which is collectively based on contemporary evidence of NDT and task-oriented partial weight-bearing treadmill gait training. Specifically, NDT included 25 min of inhibitory and facilitatory motor control exercises, core stabilization, functional training based on the concept of the neurodevelopment sequence (supine-side lying-prone rolling, quadruped, sitting, kneeling and half kneeling, transfer, plantigrade, standing, and walking training), and motor control stages (mobility, stability, and controlled mobility, and skill) [32]. The NDT-Bobath method or concept was ensured, given that all patients were treated according to the current rules of the method by the same experienced therapist. The intervention involves learning functional activities that involve sensory, perceptual, and adaptive components. Activities must involve sensorimotor experience because learning comes from movement perception [33]. Neurodevelopmental treatment is a hands-on approach that seeks to improve gross motor function in children and adults with neurological problems, thereby improving their independence in a variety of contexts [34]. It is thought that by stimulating the affected side to promote the desired muscle action, abnormal movement patterns can be corrected, and normal movement patterns conducive to performing everyday activities can be restored [34,35]. The key elements of NDT are facilitation (using sensory inputs to improve motor performance), management of compensatory motor behavior, and an overall management strategy. Kollen et al. (2009) reported that the patient must be active while the therapist assists them. The therapist assists the patient in moving, using key points of control, such as the head, shoulders, and pelvis, and guides the movement of the whole body in NDT [10]. NDT involves task-specific postures and movements. It emphasizes functional activities and participation in relevant daily life situations [35]. The main aim of NDT is to improve the quality of life of patients with neurological lesions by optimizing their level of activity and participation. Once the patient has progressed to standing, he or she engages in 5–10 min of the task-oriented partial weight-bearing treadmill gait-training. The intervention progression is adapted based on each patient’s functional level (Figure 2).

### 2.4. Statistical Analysis

Descriptive statistics included the means and standard deviation. Baseline demographics and clinical characteristics between groups were compared using the independent *t*-test for continuous variables and the chi-squared test for categorical variables. An independent *t*-test was used to determine statistical differences in the physiotherapist’s work efficiency (labor intensiveness, physical stress, or level of effort) and transmission risk of COVID-19 (social distancing, duration of contact time, and evidence of COVID-19 transmission). A 2 × 2 repeated analysis of variance (ANOVA) was used to determine statistical differences in TIS, BBS, K-MBI, MAS, and FMA scores before and after intervention between the FCR and MRR groups. A significant time × group interaction would suggest that the change between pre-and postintervention was significantly different between the control and experimental groups. Tukey’s post-hoc test was performed if interaction and main effects were observed. Statistical significance was set at *p* < 0.05. The Statistical Package for the Social Sciences (SPSS) for Windows version 25.0 (SPSS, Chicago, IL, USA) was used for statistical analysis.

## 3. Results

All participants who successfully completed the pretest, the intervention-, and the posttest were included in the analysis. Table 1 summarizes the demographic and clinical characteristics of the participants. Chi-squared and independent *t*-tests did not show significant differences in baseline demographic or clinical characteristics between the two groups (Table 1).

The independent *t*-test showed a significantly greater decrease in labor-intensity and physical stress during MRR than during FCR (*p* = 0.02 and *p =* 0.03, respectively), suggesting a greater decrease in labor demand for the physical therapist with Walkbot compared to that for conventional physical therapy (Table 2).

Moreover, the independent *t*-test showed a significantly greater decrease in social distance time, duration of contact time, and actual episode of COVID-19 contamination in the MRR than in the FCR (*p* = 0.001), indicating a greater decrease in COVID-10 contamination in the Walkbot than in conventional physical therapy during COVID-19 (Table 2).

The repeated measures ANOVA showed significant time × group (*p* = 0.001), main group for FMA (*p* = 0.001), and time effects (*p* = 0.001). Paired *t*-tests revealed significant differences in FMA scores between the MRR pre- and posttest groups (*p =* 0.001; Table 3). Tukey’s post-hoc test revealed that MRR showed a greater increase in FMA than that for FCR (*p* = 0.017).

The repeated measures ANOVA showed significant time × group (*p* = 0.001) and main group for the BBS (*p* = 0.001) and time effects (*p* = 0.001). A paired *t*-test revealed significant differences in FMA scores between the MRR pre- and posttests (*p =* 0.001) and FCR (*p* = 0.03; Table 4). Tukey’s post-hoc test showed that MRR showed a greater increase in the FMA than that for FCR (*p* = 0.003).

The repeated measures ANOVA did not demonstrate a significant time effect, between-group effect, and time × group interaction for the MAS (*p* > 0.05) in either group (Table 4).

The repeated measures ANOVA showed significant effects of both MRR and FCR on the MBI score (*p* = 0.001) and a significant difference in the MBI score between the two groups (*p* = 0.006) (Table 5). A paired t-test revealed significant differences in the MBI scores between the MRR pretest and posttest (*p =* 0.001) and FCR (*p* = 0.03) (Table 5). Tukey’s post-hoc analysis revealed that MRR showed a greater increase in the MBI than that for FCR (*p* = 0.001).

Similarly, significant effects were found in both MRR and FCR on the TIS score (*p* = 0.001), and a significant difference was also found in the TIS score between the two groups (*p* = 0.014) (Table 5). A paired t-test revealed significant differences in the TIS scores between the MRR pre- and posttests (*p =* 0.02) (Table 5). Tukey’s post-hoc analysis revealed that MRR showed a greater increase in the TIS than that for FCR (*p* = 0.004).

## 4. Discussion

The present clinical study is the first to highlight the superior positive effects of MRR over FCR alone on the physiotherapist’s work efficiency (labor intensiveness, physical stress, and level of effort), transmission risk of COVID-19, sensorimotor function, spasticity, static and dynamic balance, trunk stability, and ADLs in participants with hemiparetic stroke. Consistent with our hypothesis, the MRR group showed a significantly greater improvement than the FCR group in static and dynamic balance, trunk stability, sensorimotor function, spasticity, and ADLs. Most importantly, the MRR group showed more clinically meaningful improvements in sensorimotor recovery and functional balance and ambulation, as well as associated daily activities in the acute to chronic phase of stroke rehabilitation than the FCR group. The postquestionnaire data confirmed that that physical therapists’ work efficiency (labor intensiveness, physical stress, and level of effort) and transmission risk of COVID-19 (social distancing, duration of contact time, evidence of COVID-19 transmission) improved during robotic and manual rehabilitation.

Analysis of the postquestionnaire showed that the physical therapists’ work efficiency (labor intensiveness, physical stress, or level of effort) and transmission risk of COVID-19 (social distancing, duration of contact time, and evidence of COVID-19 transmission) were lower during robotic rehabilitation than during direct-contact rehabilitation. These important findings suggest that the robotic rehabilitation approach can be more suitable for the mitigation of labor intensiveness, physical stress, or level of efforts of the therapist, as well as for reducing the risk of potential COVID-19 transmission. Similar to the novel finding of Gracies et al. [36] in 2019, we found that MCR had less contact time (62%) than FCR, while demonstrating better clinical outcomes in terms of balance, trunk stability, spasticity, abnormal synergistic pattern, and ADLs.

The FMA analysis also demonstrated a superior positive effect of MRR (19.6%) over FCR. This result supports those of previous studies that examined the abnormal synergistic therapeutic effects of MRR in patients with hemiparetic stroke [11,37]. Kim et al. (2019) showed better recovery of abnormal synergistics following 4 weeks of RAGT (3.37%) compared to conventional physical therapy in 19 patients with hemiparetic stroke [37]. Similarly, Park et al. (2021) found improved sensorimotor function recovery following 2 weeks of RAGT (36.28%) combined with conventional physical therapy compared to conventional physical therapy alone in 20 patients with hemiparetic acute stroke [11]. Remarkably, enjoyable (virtual reality), active, repetitive locomotor movements (1000 repetitions or steps) using RAGT can facilitate agonistic activation (e.g., dorsiflexion) while reciprocally inhibiting abnormal spasticity and synergistic antagonist activation (e.g., plantarflexion) during gait.

Clinical balance analyses revealed more significant improvements in BBS scores (39.8%) in the MRR group than in the FCR group. This finding is consistent with previous robotic evidence in patients with stroke. Similarly, other Walkbot studies have shown that functional recovery of gait, balance, and mobility was enhanced after RAGT in patients with subacute hemiparetic stroke. Park et al. (2020) found enhanced dynamic and static balance (28.9%) following 2 weeks of RAGT combined with conventional physical therapy compared to conventional physical therapy alone [12]. Kim et al. (2019) reported that the end-effector robot increased BBS scores (20.4%) after 3 weeks in 58 patients with hemiparetic stroke [38]. A possible underlying mechanism for this positive improvement is that robotic locomotor training provides a sufficient number of repetitions (1800–3600 steps) with accurate proprioception inputs on the inter-coordinated hip, knee, and ankle joints, which may result in the recovery of neuroplasticity and associated functional motor skills.

Analysis of the MBI scores showed improved ADLs (13.10%) with MRR compared to FCR. This finding is consistent with studies on robotic training in patients with hemiparetic stroke that showed greater MBI improvement. Similarly, other Walkbot studies have shown that functional recovery in ADLs was enhanced after RAGT in patients with subacute hemiparetic stroke. Schwartz et al. (2009) showed improved clinical ADL measurements (10.81%) after 6 weeks of RAGT compared to those of regular gait training physical therapy in 67 patients with sub-acute stroke [39]. Similarly, Chung (2017) reported improved ambulation, mobility, balance, and ADLs (14.99%) over 4 sessions of RAGT compared with conventional physical therapy in 41 patients with stroke [40]. This finding suggests that RAGT helped patients with hemiparetic stroke overcome the fear of falling and increase their confidence on performing ADLs. Moreover, an advantage is that the Walkbot MRR produced more ‘natural’ interlimb hip–knee–ankle coordinated locomotion and a sufficient number of repetitions (up to 2000 steps) with a safe and accurate gait training protocol [11]. Compared to conventional physical therapy, such intensive, repetitive, and accurate locomotor training is sufficient to facilitate neural plasticity and associated recovery of locomotor function [41,42]. Moreover, the RAGT system is also useful for controlling posture and locomotor functions by only making minimal changes in response to the intensity of the gait training and coordination between the limbs [43,44].

Interestingly, TIS score analysis demonstrated enhanced trunk coordination and balance (49.78%) after MRR compared to FCR. This finding is consistent with that of a previous robotic locomotor study in patients recovering from a hemiparetic stroke, which also showed greater TIS improvement. Presumably, trunk coordination and trunk stability improvement may occur because the robot-assisted locomotor training system provides trunk stabilization and coordinated interlimb hip-knee-ankle joint locomotor movement guidance and associated proprioceptive and somatosensory feedback. In particular, afferent proprioceptive signals may stimulate central pattern generators in the network of motor neurons in the spinal cord and facilitate the ascending neuronal locomotor network and neuroplasticity in the sensorimotor cortex, which regulates trunk coordination and balance during locomotion [45].

A few research limitations should be considered in future studies. First, convenience sampling and randomization were performed. Nevertheless, all patients were recruited in inpatient rehabilitation during the study and underwent physical, occupational, and speech therapies and psychosocial evaluations, wherein the full spectrum of clinical care was performed among all study participants. Additionally, the lack of a follow-up evaluation can have important effects on the sustainable therapeutic effects of MRR in patients with stroke.

## 5. Conclusions

The present study demonstrated that MRR is more effective than FCR in improving abnormal synergy, balance, ADLs, and trunk stability in patients with hemiparetic stroke. Moreover, robotic-facilitated rehabilitation is beneficial for the physical therapists’ work efficiency (labor intensity, physical stress, or level of effort) and transmission risk of COVID-19. Our novel results provide clinical evidence-based insights that RAGT improves the recovery of sensorimotor function, abnormal synergy as well as balance, ADLs, and trunk stability. Moreover, it facilitates a safer environment and less labor demand than conventional stroke rehabilitation.

## Figures and Tables

**Figure 1 healthcare-10-00691-f001:**
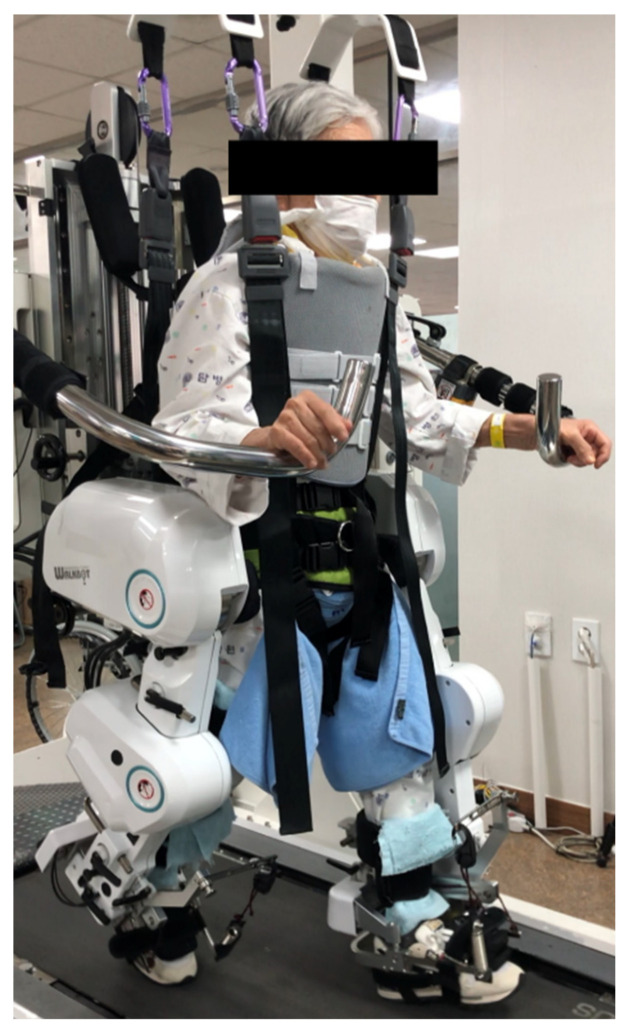
Minimal contact WALKBOT robotic rehabilitation.

**Figure 2 healthcare-10-00691-f002:**
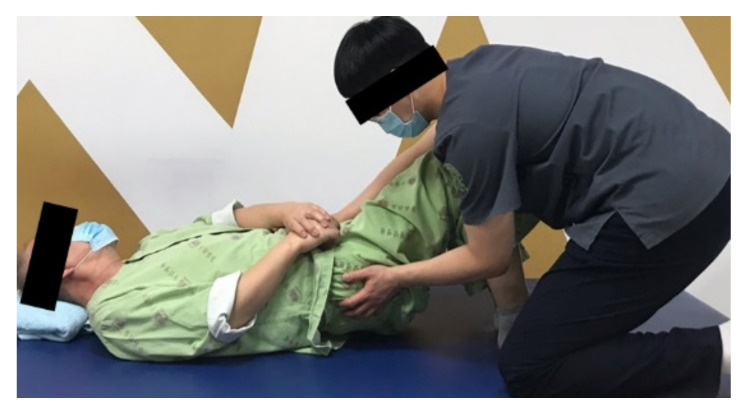
Full-contact conventional rehabilitation.

**Table 1 healthcare-10-00691-t001:** Demographic and clinical characteristics of the patients (*n* = 64).

	FCR (*n* = 32)	MRR (*n* = 32)	*p*-Value
Age (years)	63.03 ± 10.62	69.72 ± 14.72	0.07
Sex (%)			0.448
Men	20 (62.5%)	17 (53.1%)	
Women	12 (37.5%)	15 (46.9%)	
Height (cm)	164.66 ± 8.93	164.47 ± 11.73	0.943
Weight (kg)	62.56 ± 11.72	61.48 ± 12.20	0.719
Diagnosis type (%)			0.442
Ischemic	21 (65.6%)	18 (56.3%)	
Hemorrhagic	11 (34.4%)	14 (43.7%)	
Affected side (%)			0.800
Left	19 (59.4%)	18 (56.3%)	
Right	13 (40.6%)	14 (43.7%)	

FCR, full-contact conventional rehabilitation; MRR, minimal-contact robotic rehabilitation.

**Table 2 healthcare-10-00691-t002:** Postquestionnaire for the physiotherapist’s work efficiency and perceived COVID-19 transmission risk (*n* = 10).

	MRR	FCR	*p*-Value
Labor intensiveness	3.60 ± 0.84	7 ± 1.15	0.02 *
Physical stress	3.78 ± 0.38	6.11 ± 0.98	0.03 *
Social distance time (min)	23.50 ± 4.74	1.00 ± 2.10	0.001 *
Duration of contact time (min)	11.00 ± 3.94	29.00 ± 2.11	0.001 *
Perceived risk of COVID-19 transmission	1.10 ± 2.07	4.81 ± 1.15	0.001 *

FCR, full-contact conventional rehabilitation; MRR, minimal-contact robotic rehabilitation; * *p* < 0.05.

**Table 3 healthcare-10-00691-t003:** Fugl–Meyer assessment.

	MRR		FCR		*p*-Value		
	Pretest	Posttest	Pretest	Posttest	Time Effect	Between Groups	Time × Group
FMA	29.06 ± 21.71	35.97 ± 23.61	29.19 ± 31.08	30.41 ± 32.55	0.001 *	0.21	0.001 *

FCR, full-contact conventional rehabilitation; MRR, minimal-contact robotic rehabilitation; FMA, Fugl–Meyer Assessment; * *p* < 0.05.

**Table 4 healthcare-10-00691-t004:** Berg balance scale and modified Ashworth scale.

	MRR		FCR		*p*-Value		
	Pretest	Posttest	Pretest	Posttest	Time Effect	Between Groups	Time × Group
BBS	10.38 ± 9.60	17.03 ± 10.21	10.84 ± 18.00	13.47 ± 18.71	0.001 *	0.488	0.001 *
MAS	1.02 ± 0.85	0.93 ± 0.79	1.01 ± 1.55	1.01 ± 1.35	0.460	0.230	0.570

FCR, full-contact conventional rehabilitation; MRR, minimal-contact robotic rehabilitation; BBS, Berg Balance Scale; MAS, modified Ashworth Scale; * *p* < 0.05.

**Table 5 healthcare-10-00691-t005:** Modified Barthel index and Trunk Impairment Scale.

	MRR		FCR		*p*-Value		
	Pretest	Posttest	Pretest	Posttest	Time Effect	Between Groups	Time × Group
MBI	38.75 ± 17.65	48.19 ± 19.34	34.06 ± 28.97	37.91 ± 30.69	0.001 *	0.006 *	0.003 *
TIS	7.44 ± 5.29	11.53 ± 5.56	7.90 ± 8.10	8.31 ± 8.38	0.001 *	0.014 *	0.001 *

FCR, full-contact conventional rehabilitation; MRR, minimal-contact robotic rehabilitation; MBI, modified Barthel Index; TIS, Trunk Impairment Scale; * *p* < 0.05.

## Data Availability

The data presented in this study are available on request from the corresponding author.

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
