# Peer review of "Minimal Contact Robotic Stroke Rehabilitation on Risk of COVID-19, Work Efficiency and Sensorimotor Function"

_healthcare, 2022, doi:10.3390/healthcare10040691_

Round 1

Reviewer 1 Report

This paper is about a comparison of two gait rehabilitation methods for people with hemiparetic stroke: a minimal-contact robotic rehabilitation (MRR) and a full-contact conventional rehabilitation (FCR). Authors showed that the proposed MMR shows significantly better results in parameters (e.g., TIS, BBS, FMA, MBI) than FCR. It is concluded that this proposed MMR would have advantages in reducing human efforts and risks of Covid-19.

This is valuable research in the pandemic. Nevertheless, it could be better to present the proposed ideas.
At first, I recommend you change the title to "Minimal contact robotic rehabilitation ... " instead of "Untacted robotic rehabilitation ..." because I don't think the word "untacted" is neither an English word nor an academic word
Secondly. this paper needs to explain which a full-contact conventional rehabilitation method is compared with MRR. Authors should add more explanations and references.
Thirdly, the authors should explain previous literature on parameters used in this paper (e.g., TIS, BBS, FMA, MBI, etc.) in the introduction, not in the discussion.
Fourthly, authors need more study about the state of the art and explain it. For example, the authors should show if the FCR in this paper is the state of art. Otherwise, the authors should explain why the FCR is chosen for the comparison with MRR. 
Fifthly, the authors already explained robotic rehabilitation is more effective and less labor-intensive, compared to hands-on (manual) rehabilitation. So, the authors should explain if minimal-contact rototic rehabilitation (MRR) is implemented on a special condition or not. In other words, readers would wonder if the authors decided to use a special option (method) in using the walking robot, even though there are more options for the robot.
Sixthly, if the proposed method has advantages in a pandemic because of less contact, the authors need to compare the minimal contact robotic rehabilitation can be compared with the guided self- rehabilitation contact method which has less contact during rehabilitation (Gracies et. al., 2019). Or authors discuss it in the discussion section.
Seventhly, it would be nice if there is a subsection to explain more about MRR and FCR in Material and method.
Lastly, there are some minor changes that are needed.
line 40: "hands-on or manual" <- they are the same meaning...
line 46: "the main issue ... becomes ..." -> the main issues ... become ...
line 47: ".. becomes more critical during the outbreak of the COVID-19..." <- please explain the reason or put references.
line 48: " develop an effective and sustainable ..." -> " increase effectiveness and sustainability of ..." 
The adjective 'sustainable' has recently been used to mean 'going well with the environment' by 'using recycling material' or 'using renewable energy.
line 73: "..(CDW) in the hospital" <- which hospital?
line 106 - 137: it would be nice if the full name of parameters are shown one more time here.
line 145: the sub title is too long, "Post-questionnare" is enogh because the specific information appears on the next lines. 
line 155: ".. the Walkbot-G system ..." <- which company?
line 165: "Figure 1 .." <- are you explaining MRR or FCR. Does FCR also use a robotic system? It would be nice if there are two figures for both MRR and FCR.
line 195: Table 1 "MRR (n 32)" -> "MRR (n = 32)"
line 261: "It is plausible that ..." <- Do you mean it seems logical but not logical or not ture? If you would like to support the paper [28], you need to change the word "plausible" to one of the positive words, "interesting",  "remarkable" or something others. If you would like to mention active 

Author Response

Reviewer 1

This paper is about a comparison of two gait rehabilitation methods for people with hemiparetic stroke: a minimal-contact robotic rehabilitation (MRR) and a full-contact conventional rehabilitation (FCR). Authors showed that the proposed MMR shows significantly better results in parameters (e.g., TIS, BBS, FMA, MBI) than FCR. It is concluded that this proposed MMR would have advantages in reducing human efforts and risks of Covid-19.

This is valuable research in the pandemic. Nevertheless, it could be better to present the proposed ideas.
At first, I recommend you change the title to "Minimal contact robotic rehabilitation ... " instead of "Untacted robotic rehabilitation ..." because I don't think the word "untacted" is neither an English word nor an academic word

Authors response: The title was revised.

Secondly. this paper needs to explain which a full-contact conventional rehabilitation method is compared with MRR. Authors should add more explanations and references.
Thirdly, the authors should explain previous literature on parameters used in this paper (e.g., TIS, BBS, FMA, MBI, etc.) in the introduction, not in the discussion.

Authors response: This was revised (Lines 38-40).

Fourthly, authors need more study about the state of the art and explain it. For example, the authors should show if the FCR in this paper is the state of art. Otherwise, the authors should explain why the FCR is chosen for the comparison with MRR. 

Authors response: FCR is not the state-of-the-art approach, but a conventional rehabilitation which includes manual therapeutic exercises using the collective evidence in the NDT and task-oriented partial weight-bearing treadmill gait training approaches. This was revised (Lines 38-40).

Fifthly, the authors already explained robotic rehabilitation is more effective and less labor-intensive, compared to hands-on (manual) rehabilitation. So, the authors should explain if minimal-contact robotic rehabilitation (MRR) is implemented on a special condition or not. In other words, readers would wonder if the authors decided to use a special option (method) in using the walking robot, even though there are more options for the robot.

Authors response: The reviewer’s point is well taken. As the reviewer indicated, this was clarified in the introduction section of the revised manuscript to read: “Therefore, the primary objective of the present study was to compare the risk on vulnerable for COVID-19 including transmission risk of COVID-19 (social distancing, duration of contact time, and evidence of COVID-19 transmission) and physiotherapist’s work efficiency (labor intensiveness, physical stress, or level of effort). The secondary objective was to ascertain the effects of MRR and FCR compare on the static and dynamic balance, trunk stability, ADLs, and spasticity in a larger sample of patients with hemiparetic stroke. We hypothesized that MRR would improve clinical indices on the risk on vulnerable for COVID-19, Trunk Impairment Scale (TIS), Berg Balance Scale (BBS), modified Ashworth Scale (MAS), Fugl-Meyer assessment (FMA), and modified Barthel Index (MBI) when compared to those of FCR conventional physiotherapy.” (Lines 61-70)

Sixthly, if the proposed method has advantages in a pandemic because of less contact, the authors need to compare the minimal contact robotic rehabilitation can be compared with the guided self- rehabilitation contact method which has less contact during rehabilitation (Gracies et. al., 2019). Or authors discuss it in the discussion section.

Authors response: As the reviewers suggested, this content was added in the revised manuscript to read: “Similar to Gracies and colleagues (2019) novel finding [33], we MCR had less contact time (62%) when compared to FCR while demonstrating better clinical outcomes in balance, trunk stability, spasticity, abnormal synergistic pattern and ADLs.” (Lines 306-309)

Seventhly, it would be nice if there is a subsection to explain more about MRR and FCR in Material and method.

Authors response: This was revised (Lines 165-177; 183-205)

Lastly, there are some minor changes that are needed.
line 40: "hands-on or manual" <- they are the same meaning...

Authors response: This was revised (Line 44)

line 46: "the main issue ... becomes ..." -> the main issues ... become ...

Authors response: This was revised (Line 44)

line 47: ".. becomes more critical during the outbreak of the COVID-19..." <- please explain the reason or put references.

Authors response: This was revised (Line 45)

line 48: " develop an effective and sustainable ..." -> " increase effectiveness and sustainability of ..." 
The adjective 'sustainable' has recently been used to mean 'going well with the environment' by 'using recycling material' or 'using renewable energy.

Authors response: This was revised (Line 46)

line 73: "..(CDW) in the hospital" <- which hospital?

Authors response: This was revised (Line 77)

line 106 - 137: it would be nice if the full name of parameters are shown one more time here.

Authors response: This was revised

line 145: the sub title is too long, "Post-questionnare" is enogh because the specific information appears on the next lines. 

Authors response: This was revised (Line 150)

line 155: ".. the Walkbot-G system ..." <- which company?

Authors response: This was revised (Line 159)

line 165: "Figure 1 .." <- are you explaining MRR or FCR. Does FCR also use a robotic system? It would be nice if there are two figures for both MRR and FCR.

Authors response: This was revised (Figure 2)

line 195: Table 1 "MRR (n 32)" -> "MRR (n = 32)"

Authors response: This was revised

line 261: "It is plausible that ..." <- Do you mean it seems logical but not logical or not ture? If you would like to support the paper [28], you need to change the word "plausible" to one of the positive words, "interesting",  "remarkable" or something others. If you would like to mention active 

Authors response: This was revised (Line 319)

Reviewer 2 Report

The paper presents the advantages of using a rehabilitation robotic solution called Walkbot to mitigate the issues surrounding conventional hands-on physical therapy during the COVID-19 global pandemic. The authors compare the effects of minimal-contact robotic rehabilitation and full-contact conventional rehabilitation on static and dynamic balance, trunk stability, activities of daily living, spasticity and cognition changes in patients with hemiparetic stroke. The presented results provide clinical evidence that robot-assisted locomotor training helps maximize the recovery of sensorimotor function, abnormal synergy, balance, ADLs, and trunk stability, and facilitates a safer environment and less labor demand than conventional stroke rehabilitation.

The paper brings several contributions to the research oriented on establishing the efficiency of rehabilitation methods after stroke using robotic solutions. The presented analyze determines the paper to be considered as a good guide in support of the researchers in the field.

My recommendation for the authors is to submit a light revised version, improving the following aspects:

  • The title is too long. Try to change it into a more compact one, keeping the main idea regarding the advantages that come from the minimal-contact robotic rehabilitation
  • Underline the originality of the paper in a more clear manner since the superiority of robotic solutions have been emphasized in many research works due to the fact that data acquired by the rehabilitation robotic systems and the applications associated to them that automatically show the real-time rehabilitation progress have proved clear advantages many times.

Author Response

The paper presents the advantages of using a rehabilitation robotic solution called Walkbot to mitigate the issues surrounding conventional hands-on physical therapy during the COVID-19 global pandemic. The authors compare the effects of minimal-contact robotic rehabilitation and full-contact conventional rehabilitation on static and dynamic balance, trunk stability, activities of daily living, spasticity and cognition changes in patients with hemiparetic stroke. The presented results provide clinical evidence that robot-assisted locomotor training helps maximize the recovery of sensorimotor function, abnormal synergy, balance, ADLs, and trunk stability, and facilitates a safer environment and less labor demand than conventional stroke rehabilitation.

The paper brings several contributions to the research oriented on establishing the efficiency of rehabilitation methods after stroke using robotic solutions. The presented analyze determines the paper to be considered as a good guide in support of the researchers in the field.

My recommendation for the authors is to submit a light revised version, improving the following aspects:

  • The title is too long. Try to change it into a more compact one, keeping the main idea regarding the advantages that come from the minimal-contact robotic rehabilitation

Authors response: The title was revised.

  • Underline the originality of the paper in a more clear manner since the superiority of robotic solutions have been emphasized in many research works due to the fact that data acquired by the rehabilitation robotic systems and the applications associated to them that automatically show the real-time rehabilitation progress have proved clear advantages many times.

Authors response: The title was revised.

Round 2

Reviewer 2 Report

The revision was substantial and good.

I am satisfied with the author's responses.

Based on them I recommend this paper to be considered for publication.